# Long Covid symptoms and diagnosis in primary care: A cohort study using structured and unstructured data in The Health Improvement Network primary care database

**Anoop D. Shah**[1,2]*, **Anuradhaa Subramanian**[3], **Jadene Lewis**[1], **Samir Dhalla**[4], **Elizabeth Ford**[5], **Shamil Haroon**[3], **Valerie Kuan**[1], **Krishnarajah Nirantharakumar**[3]

**1** Institute of Health Informatics, University College London, London, United Kingdom, **2** NIHR University College London Hospitals Biomedical Research Centre, University College London Hospitals NHS Trust, London, United Kingdom, **3** Institute of Applied Health Research, University of Birmingham, Birmingham, United Kingdom, **4** The Health Improvement Network Ltd., London, United Kingdom, **5** Brighton and Sussex Medical School, Brighton, United Kingdom

* anoop@doctors.org.uk

**Data Availability Statement:** Data cannot be shared publicly because of patient confidentiality; the study uses individual patient electronic health

## Abstract

### Background

Long Covid is a widely recognised consequence of COVID-19 infection, but little is known about the burden of symptoms that patients present with in primary care, as these are typically recorded only in free text clinical notes.

### Aims

To compare symptoms in patients with and without a history of COVID-19, and investigate symptoms associated with a Long Covid diagnosis.

### Methods

We used primary care electronic health record data until the end of December 2020 from The Health Improvement Network (THIN), a Cegedim database. We included adults registered with participating practices in England, Scotland or Wales. We extracted information about 89 symptoms and 'Long Covid' diagnoses from free text using natural language processing. We calculated hazard ratios (adjusted for age, sex, baseline medical conditions and prior symptoms) for each symptom from 12 weeks after the COVID-19 diagnosis.

### Results

We compared 11,015 patients with confirmed COVID-19 and 18,098 unexposed controls. Only 20% of symptom records were coded, with 80% in free text. A wide range of symptoms were associated with COVID-19 at least 12 weeks post-infection, with strongest associations for fatigue (adjusted hazard ratio (aHR) 3.46, 95% confidence interval (CI) 2.87, 4.17), shortness of breath (aHR 2.89, 95% CI 2.48, 3.36), palpitations (aHR 2.59, 95% CI 1.86, 3.60), and phlegm (aHR 2.43, 95% CI 1.65, 3.59). However, a limited subset of symptoms

record data without consent. Data are available from The Health Improvement Network Ltd. (THIN) for researchers who meet the criteria for access to confidential data (contact via info@the-health-improvement-network.com). Approval will be required from the THIN Advisory Committee, and in the case of unstructured text, approval will also be required from the Health Research Authority Confidentiality Advisory Group.

**Funding:** This work was supported by Health Data Research UK, which receives its funding from the UK Medical Research Council, Engineering and Physical Sciences Research Council, Economic and Social Research Council, Department of Health and Social Care (England), Chief Scientist Office of the Scottish Government Health and Social Care Directorates, Health and Social Care Research and Development Division (Welsh Government), Public Health Agency (Northern Ireland), British Heart Foundation, and the Wellcome Trust. This study was supported by the National Institute for Health Research (NIHR) CONVALESCENCE grant (COV-LT-0009). ADS is funded by a postdoctoral fellowship from THIS Institute, NIHR (AI_AWARD01864 and COV-LT-0009), UKRI (Horizon Europe Guarantee for DataTools4Heart) and British Heart Foundation Accelerator Award (AA/18/6/24223). VK is supported by the UKRI/NIHR Strategic Priorities Award in Multimorbidity Research (MR/V033867/1) for the Multimorbidity Mechanism and Therapeutics Research Collaborative. EF is supported by the NIHR Applied Research Collaboration Kent Surrey and Sussex (grant number NIHR200179). The funders had no role in study design, data collection and analysis, decision to publish, or preparation of the manuscript.

**Competing interests:** ADS has received research grants from THIS Institute, NIHR, UKRI and BHF. VK has received a research grant from UKRI. EF has received research grants from NIHR. KN has been awarded research grants from NIHR, UKRI/MRC, Kennedy Trust for Rheumatology Research, Health Data Research UK, Wellcome Trust, European Regional Development Fund, Institute for Global Innovation, Boehringer Ingelheim, Action Against Macular Degeneration Charity, Midlands Neuroscience Teaching and Development Funds, South Asian Health Foundation, Vifor Pharma, College of Police, and CSL Behring, all payments were made to his academic institution. KN has received consulting fees from BI, Sanofi, CEGEDIM, MSD and holds a leadership/fiduciary role with NICST, a charity and OpenClinical, a Social Enterprise. None of the authors has involvement in products or patents relevant to this

were recorded within 7 days prior to a Long Covid diagnosis in more than 20% of cases: shortness of breath, chest pain, pain, fatigue, cough, and anxiety / depression.

## Conclusions

Numerous symptoms are reported to primary care at least 12 weeks after COVID-19 infection, but only a subset are commonly associated with a GP diagnosis of Long Covid.

## Introduction

Long-term symptoms are a well recognised consequence of COVID-19 infection [1–12], and there is a need to better understand the condition in order to improve diagnosis and care [13, 14]. Previous studies on Long Covid symptoms have used a variety of methods, each with strengths but also limitations. Studies based on patient reports [5, 7, 15] or symptom tracker apps [16, 17] provide a detailed picture of symptom experiences, but are subject to selection bias [18] and often lack a comparator group. Existing longitudinal cohort studies allow symptom prevalence post COVID-19 to be compared with patients who have not had COVID-19, but these have small numbers of patients [19].

Many patients with ongoing symptoms post COVID-19 present to their general practitioners (GPs), and there is currently little information on which symptoms patients attend with [20], and the basis on which GPs assign a diagnostic label of 'Long Covid'. There have been numerous studies on clinical characteristics of Long Covid using electronic health record data [6, 8–12], but few in the UK general practice setting [14]. Primary care data have already been shown to be invaluable for understanding population risk, morbidity and mortality due to COVID-19 [21, 22], and have been used to study coded Long Covid diagnoses [19, 23]. However, symptoms are typically not recorded in a structured way in primary care records [24, 25].

We aimed to address this gap using natural language processing to extract information about symptoms recorded in primary care consultations [26], thus overcoming the limitation of coded data [14, 27]. Our study is based on data to the end of December 2020, i.e. the first wave of COVID-19 infections in a predominantly unvaccinated population. Our primary objectives were to (1) describe the long term profile of symptoms as recorded in general practice for patients with COVID-19, (2) compare symptoms in patients with and without a history of COVID-19 infection, (3) identify symptoms associated with a GP diagnosis of 'Long Covid'. We also conducted an exploratory analysis of clustering of Long Covid symptoms and risk factors for Long Covid.

## Methods

### Data source

We used primary care electronic health record data from patients in England, Scotland, or Wales registered for at least 1 year with general practices contributing to The Health Improvement Network (THIN), a Cegedim Database [28]. We used structured data (including diagnoses and symptoms coded using the Read Clinical Terminology), and unstructured data (free text clinical notes in the primary care record). The study period was 1 December 2019 to 31 December 2020, and free text clinical notes from this time period were analysed. Structured data up to and including 31 December 2020 were analysed in this study; this included data

study. This does not alter our adherence to PLOS ONE policies on sharing data and materials.

prior to the start of the study period (1 December 2019) such as historic diagnoses for baseline characterisation of patients.

## Study population

We defined four fixed, non-overlapping patient cohorts among adult patients registered with a general practice contributing to THIN during the study period. Because of the need to use data from free text to help classify patients, and restrictions on access to the full THIN database, cohort selection was carried out in a stepwise manner (Fig 1). In the first step, exposed cohorts (confirmed or suspected COVID-19) were selected based on Read codes for suspected or confirmed COVID-19 (list of Read terms in S1 Table). Each patient's index date was defined as

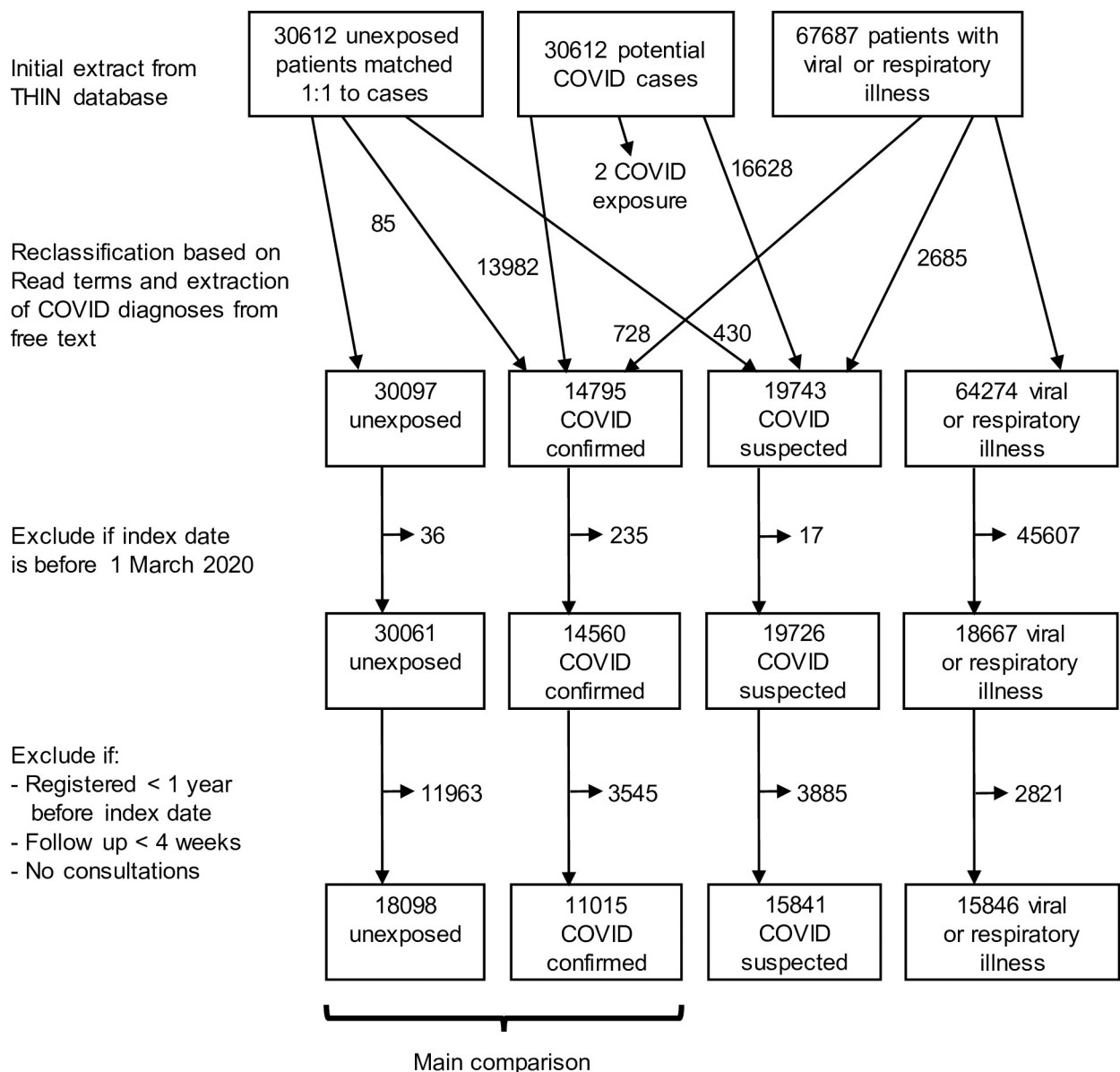

**Fig 1. CONSORT diagram showing selection of patients in each category.**

the date of their earliest record of COVID-19. Unexposed patients were then selected using a 1:1 matching algorithm to patients with confirmed or suspected COVID-19, in order to assign index dates to unexposed patients and ensure a similar distribution of key demographics between the cohorts. The matching variables were practice, age (+/- 3 years) and sex. We also defined a 'possibly exposed' cohort with Read terms for non-specific viral or respiratory illness (S2 Table). These were patients who had symptoms compatible with COVID-19 but no formal COVID-19 diagnosis. The index date for this cohort was defined as the earliest record of a non-specific viral or respiratory illness within the study period.

We extracted free text notes for these initial cohorts and then refined the cohort selection based on COVID-19 diagnoses or test results extracted from the free text. We amended the index date if appropriate based on information in the free text; for example if a patient had a free text diagnosis of COVID-19 prior to their coded entry their index date was brought forward. If patients had a confirmed COVID-19 diagnosis they were included in the confirmed COVID-19 cohort regardless of other diagnoses recorded, and if patients had a suspected COVID-19 but not confirmed COVID-19 they were included in the suspected COVID-19 cohort whether or not they had a viral or respiratory illness recorded.

We excluded patients registered for less than 1 year prior to the index date, or with less than 4 weeks ongoing registration after the index date, or if there were no consultations recorded at all (Fig 1).

## Data extraction

For each patient, we extracted demographic details (age, sex, and ethnicity), lifestyle information (smoking status) and clinical measurements (body mass index) from their primary care record. Socioeconomic information was available at practice level (index of multiple deprivation, IMD quintile). We extracted information about symptoms before and after COVID-19 diagnosis, whether the patient was hospitalised within 14 days prior to 28 days after their index date, and the number of consultations in the year before the index date.

We extracted information from free text using a rule-based named entity recognition and linking algorithm called the Freetext Matching Algorithm (FMA), which has previously been validated on primary care free text [24, 27]. FMA maps information about symptoms, hospitalisation and diagnoses to Read terms, and includes rule-based methods for detecting negation, uncertainty and relevance. We used FMA to supplement the structured data for classifying cases and controls, ascertaining if a patient was hospitalised, and whether they reported specific symptoms. We manually validated a sample of texts containing extracted items of interest (see Supplementary Methods in S1 Text). The authors did not have access to information that could identify individual participants during or after data collection.

## Recording of symptoms

We used similar definitions of symptoms to a recent study by Subramanian et al. using the Clinical Practice Research Datalink (CPRD) Aurum database [14], who studied 115 symptoms using coded clinical data. Given the smaller patient population in our study, we combined some symptoms to produce a final list of 89 symptoms, and present the main results for the 30 most commonly recorded symptoms.

As an initial assessment of overall symptom burden, we calculated odds ratios for symptom recording by patient category in 4-week periods after the index date compared to a reference period 8–12 weeks before the index date.

## Comparison of symptoms in patients with and without COVID-19

We used Cox proportional hazards models to compare recording of symptoms (as clinical codes or free text) in exposed patients (separately for those with confirmed COVID-19, suspected COVID-19, or non-specific viral or respiratory illnesses), and unexposed controls. We analysed data for each symptom separately. The primary analysis was for the time period starting 12 weeks after the index date, i.e. the cut-off beyond which persistent symptoms may contribute to a Long COVID diagnosis according to World Health Organization (WHO) criteria [29]. Patients were followed up until their first occurrence of the symptom of interest, or censored on the earliest of end of study period, last collection date, date of death or transfer out of the practice. Hazard ratios were adjusted for age, sex, age/sex interaction, number of consultations in the year before the index date, number of days on which any symptom was recorded 1–3 months before the index date, recording of the specific symptom 1–3 months before the index date, ethnicity, smoking, body mass index and a generated propensity score for acquiring COVID-19 infection which incorporates prior diagnoses according to the SNOMED CT hierarchy [30] (see Supplementary Methods in S1 Text). We incorporated the propensity score as an additional variable rather than using weighting methods to avoid issues with extreme weights [31]. We stratified baseline hazard functions by general practice to account for variation between practices. Missing values of ethnicity, smoking and body mass index were classed as a separate category for analysis.

We carried out subgroup analyses by time period, sex, age group and nation, and sensitivity analyses using different levels of adjustment or limited to coded data only.

## Symptoms associated with a GP diagnosis of Long Covid

We sought to investigate the basis on which GPs were suspecting or making a diagnosis of Long Covid. For patients in the 'confirmed COVID-19' category with a GP diagnosis of confirmed or suspected Long Covid at least 12 weeks after the index date, we calculated the proportion with each symptom recorded in the prior week.

## Clustering and risk factors for Long Covid

For the latent class analysis (LCA) and risk factor analysis we operationalised 'Long Covid' as the presence of at least one symptom included in the WHO case definition of post COVID-19 condition [29] at least 12 weeks after the index date, among patients with confirmed COVID-19. We did not attempt to assign a 'Long Covid' diagnosis as this was not possible using the available data, instead we aimed to identify a patient population with symptoms that may be consistent with Long Covid for the purpose of the clustering analysis, similar to the approach used by Subramanian et al. [14]. We characterised patients by the presence or absence of symptoms recorded in the 3 months after the first WHO symptom, and excluded patients without a full 3-month follow up period after this date (to avoid the influence of follow-up duration on symptom recording). We used an elbow plot to identify the optimum number of clusters.

We used Cox models to investigate associations of Long Covid with age, sex, number of consultations, ethnicity, smoking, body mass index, hospitalisation, practice-level deprivation quintile, and symptoms prior to the index date. We also carried out the risk factor analysis using GP diagnosis of suspected or confirmed Long Covid as the outcome. We carried out analyses using the R statistical system (version 4.1) [32], using the survival, poLCA, and glmnet packages. Our analysis code is included in S1 File.

### Ethics

The THIN database has overarching Health Research Authority ethical approval for observational research (20/SC/0011, Jan 2020). Our study protocol was approved by the North East–Tyne & Wear South Research Ethics Committee (20/NE/0209). The need for informed consent was waived by the ethics committee. Use of identifiable patient data in England and Wales was permitted by the Covid-19 –Notice under Regulation 3(4) of the Health Service (COPI, Control of Patient Information) Regulations 2002. We confirmed with the Scottish Patient and Public Benefit Panel that free text data from primary care could be used for research with appropriate data sharing agreements in place. The authors had access to only deidentified patient information, and did not have access to information that could identify individual participants during or after data collection.

## Results

### Study population

We included 11,015 confirmed COVID-19 cases, 15,841 suspected COVID-19 cases, 15,846 possibly exposed patients (with a viral or respiratory illness) and 18,098 unexposed controls. The initial search criteria selected 30,612 patients with confirmed or suspected COVID-19, and 3930 patients were reclassified based on information extracted from the free text (Fig 1). 63% of the study participants (38,407 / 60,800) were female, and the mean age was 52 years (Table 1). Roughly equal number of patients were from each of Scotland (21,133), England (19,123) and Wales (20,544). Almost 90% of those with ethnicity recorded (89.6%, 25,287 / 28,236) were of White ethnicity (Table 1). Only a small proportion of patients had received a COVID-19 vaccination prior to the index date. Patients were followed up for a median 136 days (IQR 59, 246).

### Recording of symptoms

The majority of symptom mentions in the general practice records (80%) were in the free text, with only 20% in structured data, although this varied by symptom (S3 Table). Manual validation of text samples showed precision of 85–97% on the majority of information extraction tasks, with no significant difference in precision between symptoms from COVID-19 cases (261 / 294, 88.8%) and controls (246 / 289, 85.1%), $p = 0.24$ by proportion test (see Supplementary Results in S1 Text). Inter-rater reliability of the manual annotators was good for symptoms (unweighted kappa 0.75, 95% CI 0.66, 0.83) but moderate overall (weighted kappa 0.54, 95% CI 0.48, 0.61).

There was a persistently elevated level of symptom recording for at least 9 months after the index date for confirmed and suspected COVID-19 cases (greater for confirmed cases) (Fig 2).

### Comparison of symptoms in patients with and without COVID-19

A wide range of symptoms were associated with COVID-19 beyond 12 weeks from infection (Fig 3 and S2 Fig), with the strongest associations for fatigue (adjusted hazard ratio (aHR) 3.46, 95% confidence interval (CI) 2.87, 4.17), shortness of breath (aHR 2.89, 95% CI 2.48, 3.36), palpitations (aHR 2.59, 95% CI 1.86, 3.60), and phlegm (aHR 2.43, 95% CI 1.65, 3.59).

Anosmia, fever and headache were more strongly associated with COVID at earlier time points (Fig 4), fitting the expected clinical picture. Associations observed with coded data were similar to those including free text, but the number of events was smaller so estimates were less precise (Fig 5). Associations of the majority of symptoms were stronger for patients with confirmed COVID-19 than those with suspected COVID-19 or nonspecific viral illnesses (Fig 6). Crude associations

**Table 1. Baseline characteristics of patients by cohort.**

| | Unexposed | Confirmed COVID | Suspected COVID | Viral or respiratory illness |
|---|---|---|---|---|
| Number of patients | 18098 | 11015 | 15841 | 15846 |
| Female, n (%) | 11408 (63.0%) | 6782 (61.6%) | 10093 (63.7%) | 10124 (63.9%) |
| Age in years, mean (SD) | 53.6 (18.9) | 50.5 (18.1) | 52.1 (18.9) | 51.5 (19.4) |
| Smoking status | | | | |
| Never | 8269 (47.5%) | 5545 (52.6%) | 6424 (41.8%) | 6460 (42.0%) |
| Past | 5911 (33.9%) | 3736 (35.4%) | 5448 (35.5%) | 5496 (35.7%) |
| Current | 3239 (18.6%) | 1266 (12.0%) | 3479 (22.7%) | 3421 (22.2%) |
| Missing smoking status | 679 (3.8%) | 468 (4.2%) | 490 (3.1%) | 469 (3.0%) |
| Ethnicity | | | | |
| White | 7915 (89.9%) | 4589 (89.5%) | 6601 (88.9%) | 6182 (89.9%) |
| Black | 245 (2.8%) | 116 (2.3%) | 217 (2.9%) | 139 (2.0%) |
| South Asian | 350 (4.0%) | 251 (4.9%) | 374 (5.0%) | 381 (5.5%) |
| Mixed | 81 (0.9%) | 46 (0.9%) | 98 (1.3%) | 51 (0.7%) |
| Other | 214 (2.4%) | 126 (2.5%) | 138 (1.9%) | 122 (1.8%) |
| Missing ethnicity | 9293 (51.3%) | 5887 (53.4%) | 8413 (53.1%) | 8971 (56.6%) |
| BMI category | | | | |
| <18.5 | 389 (2.5%) | 211 (2.2%) | 435 (3.1%) | 411 (2.9%) |
| 18.5–25 | 5190 (33.0%) | 2584 (27.0%) | 4115 (29.2%) | 4190 (29.8%) |
| 25–30 | 5280 (33.6%) | 3189 (33.3%) | 4324 (30.6%) | 4534 (32.2%) |
| 30–40 | 4117 (26.2%) | 2910 (30.4%) | 4230 (30.0%) | 3990 (28.3%) |
| 40+ | 761 (4.8%) | 686 (7.2%) | 1010 (7.2+) | 956 (6.8%) |
| Missing BMI | 2361 (13.0%) | 1435 (13.0%) | 1727 (10.9%) | 1765 (11.1%) |
| Most deprived quintile, n (%) | 5319 (29.4%) | 3602 (32.7%) | 4111 (26.0%) | 3962 (25.0%) |
| Median (IQR) number of GP consultations in year prior to index date (excluding index consultation) | 5 (2, 10) | 6 (2, 12) | 9 (4, 16) | 8 (4, 15) |
| Number of days with symptom mentions 1–3 months before index date | | | | |
| 0 | 13207 (73.0%) | 7273 (66.0%) | 8535 (53.9%) | 9191 (58.0%) |
| 1 | 1288 (7.1%) | 826 (7.5%) | 1382 (8.7%) | 1273 (8.0%) |
| 2 | 1035 (5.7%) | 733 (6.7%) | 1275 (8.0%) | 1155 (7.3%) |
| 3+ | 2568 (14.2%) | 2183 (19.8%) | 4649 (29.3%) | 4227 (26.7%) |
| Received at least one dose of COVID-19 vaccination before index date | 35 (0.2%) | 47 (0.4%) | 15 (0.1%) | 12 (0.1%) |
| Hospitalised within 28 days of index date | 196 (1.1%) | 1518 (13.8%) | 1887 (11.9%) | 902 (5.7%) |
| UK nation | | | | |
| England | 5759 (31.8%) | 2060 (18.7%) | 6687 (42.2%) | 4617 (29.1%) |
| Scotland | 6600 (36.5%) | 5163 (46.9%) | 4407 (27.8%) | 4963 (31.3%) |
| Wales | 5739 (31.7%) | 3792 (34.4%) | 4747 (30.0%) | 6266 (39.5%) |

SD, standard deviation; BMI, body mass index; IQR, interquartile range

were stronger than the adjusted estimates in the main analysis (S3 Fig). The majority of associations were similar across subgroups of age (S4 Fig), sex (S5 Fig) and nation (S6 Fig).

## Symptoms associated with a GP diagnosis of Long Covid

No patients had a coded diagnosis of post COVID condition (Long Covid), but there were 818 records of suspected or confirmed Long Covid in the free text among the cohort (553 unique

## Odds ratio for symptom mention over time

**Fig 2. Timeline of symptom mentions.** Odds ratio for record of any coded or free text symptom in a 4-week period, compared to 8–12 weeks prior to index date, by cohort (confirmed or suspected COVID, viral/respiratory illness or control).

patients). Among patients with confirmed Covid at least 12 weeks prior, 103 individuals (0.9%) had a free text entry for confirmed or suspected Long Covid. The most common symptoms recorded in the week prior to a Long Covid diagnosis were pain (68.3%; 95% CI 62.5%, 73.8%) shortness of breath (66.2%; 95% CI 60.3%, 71.7%) and fatigue (57.9%; 95% CI 51.9%, 63.8%) (Fig 3). Chest pain, cough, and anxiety / depression were also recorded in over 20% of cases.

On the other hand, gastrointestinal symptoms were rarely recorded in the week prior to a Long Covid diagnosis, despite being common (the total number of events for nausea / vomiting was 588, almost as high as 606 for fatigue) and strongly associated with COVID-19 (aHR 1.76 for nausea / vomiting, 95% CI 1.45, 2.13) (Fig 3). Wheezing, limb swelling, palpitations / tachycardia, phlegm, and muscle pain were also infrequently recorded in the week prior to a Long Covid diagnosis, despite a strong association with COVID-19 (aHR > 1.8) (Fig 3).

### Clustering and risk factors for Long Covid

Elbow plots of goodness of fit measures showed that a two class LCA model provided a best fit to the data (S7 Fig), with the classes fitting descriptions of high or low symptom burden rather than distinctly different sets of symptoms (S4 Table). We also present a three class model for comparison with a previous study using CPRD Aurum [14] (S5 Table).

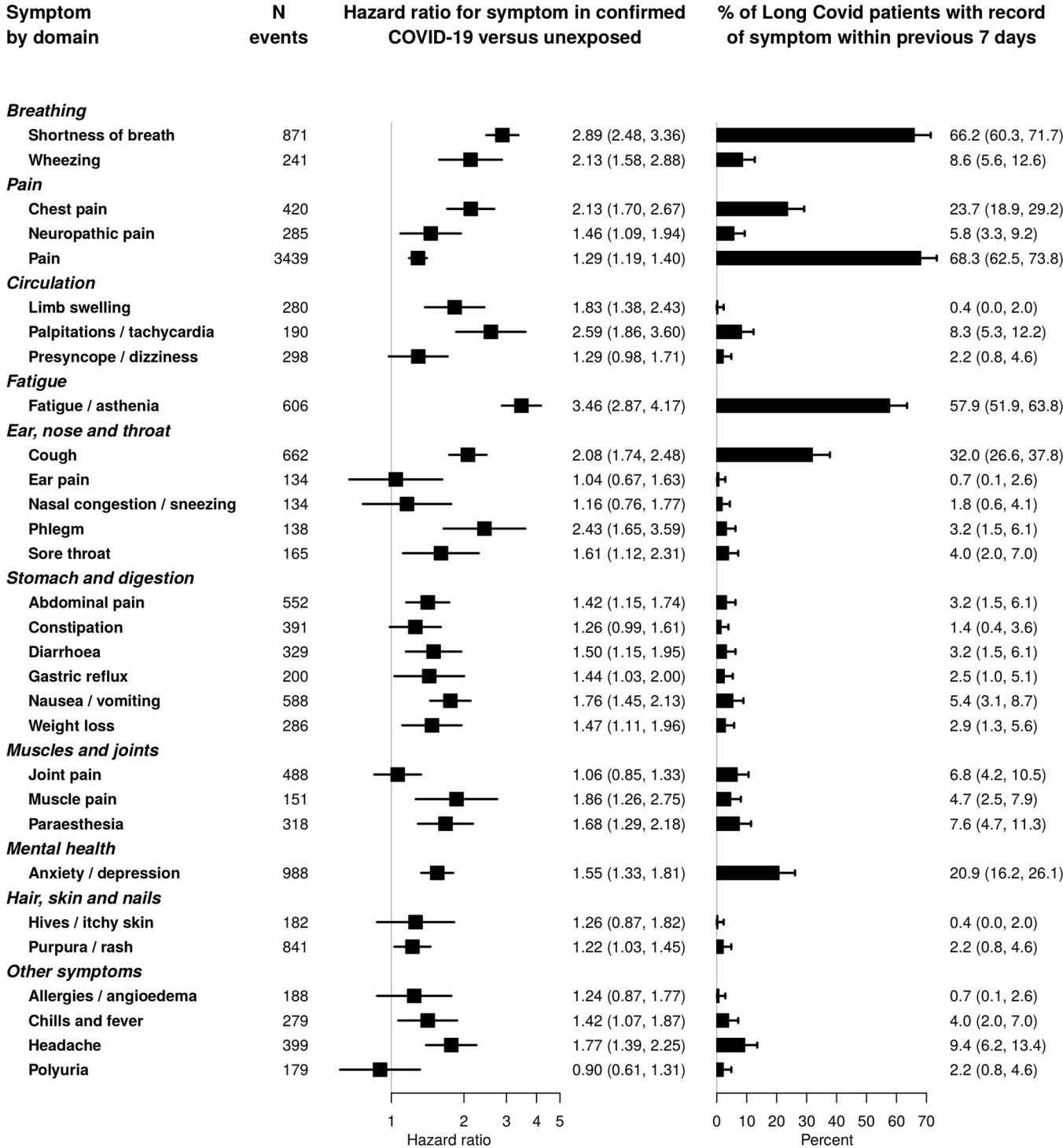

**Fig 3. Association of symptoms with prior COVID-19 infection and GP diagnosis of Long Covid.** Hazard ratios for association of symptoms with previous COVID-19 infection after 12 weeks for 30 most common symptoms, and proportion of Long Covid patients (according to suspected or confirmed GP diagnosis of Long Covid) with the symptom recorded in the preceding 7 days. Hazard ratios were adjusted for age, sex, age/sex interaction, number of consultations in the year before the index date, number of symptom days 1–3 months before the index date, recording of the specific symptom 1–3 months before the index date, ethnicity, smoking, body mass index and a generated propensity score for acquiring COVID-19 infection, and stratified by general practice.

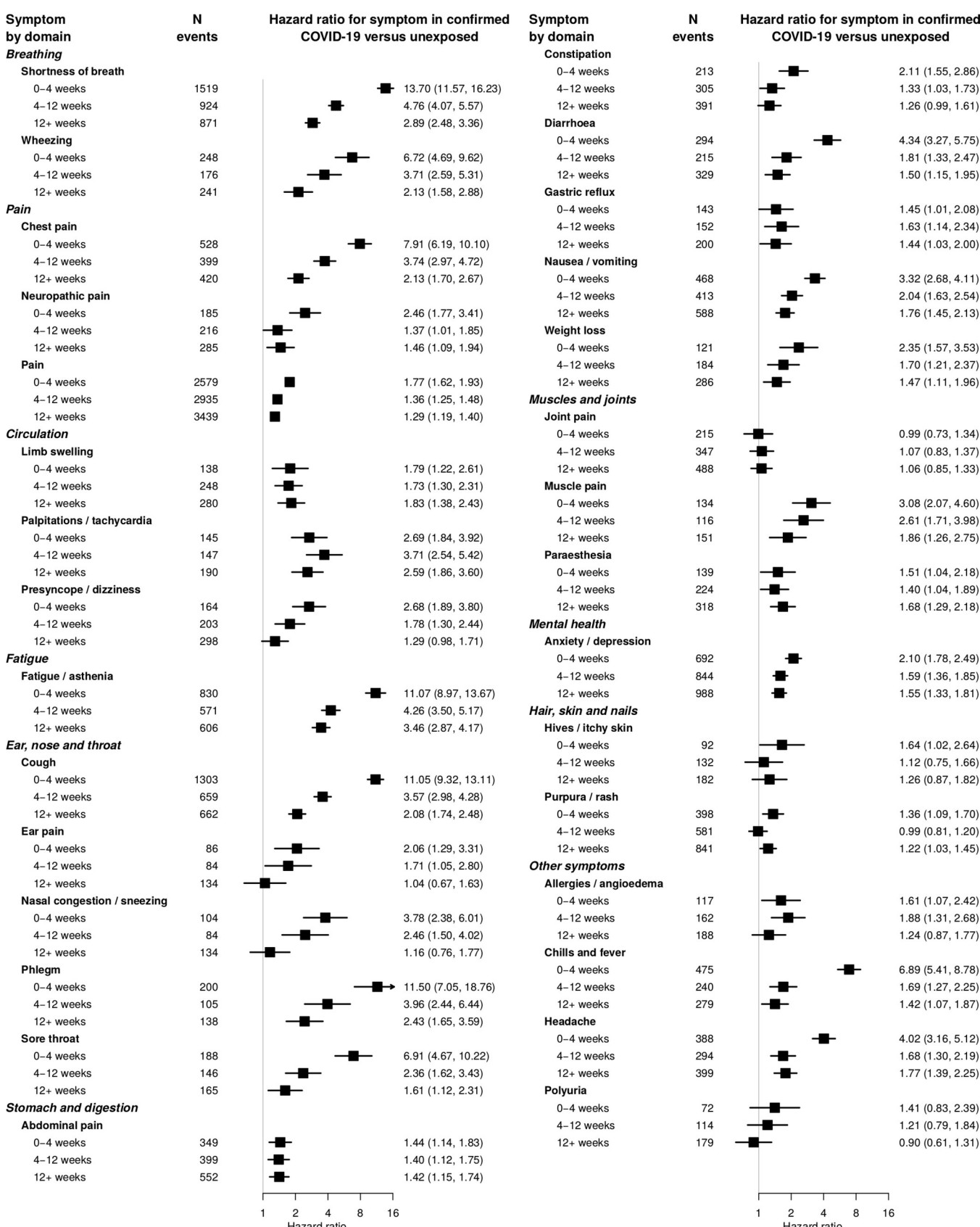

**Fig 4. Association of symptoms with prior COVID-19 infection by time.** Hazard ratios for association of symptoms with previous COVID infection by time. Hazard ratios were adjusted for age, sex, age/sex interaction, number of consultations in the year before the index date, number of symptom days 1–3 months before the index date, recording of the specific symptom 1–3 months before the index date, ethnicity, smoking, body mass index and a generated propensity score for acquiring COVID-19 infection, with the baseline hazard function stratified by general practice.

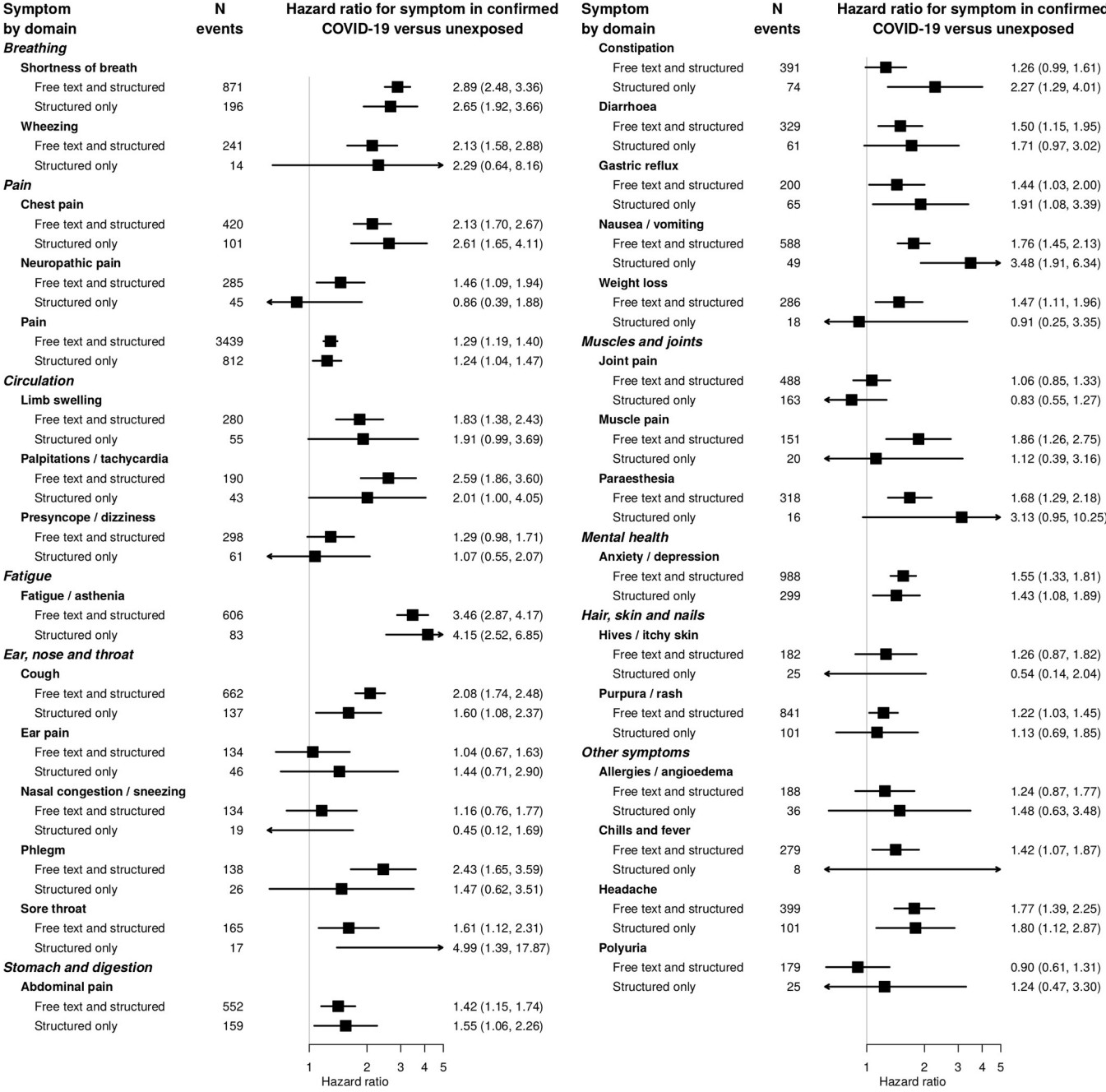

**Fig 5. Association of symptoms with prior COVID-19 infection by source of symptom data.** Hazard ratios for association of symptoms with previous COVID infection after 12 weeks, by source of symptom data (free text or structured data). Hazard ratios were adjusted for age, sex, age/sex interaction, number of consultations in the year before the index date, number of symptom days 1–3 months before the index date, recording of the specific symptom 1–3 months before the index date, ethnicity, smoking, body mass index and a generated propensity score for acquiring COVID-19 infection, with the baseline hazard function stratified by general practice.

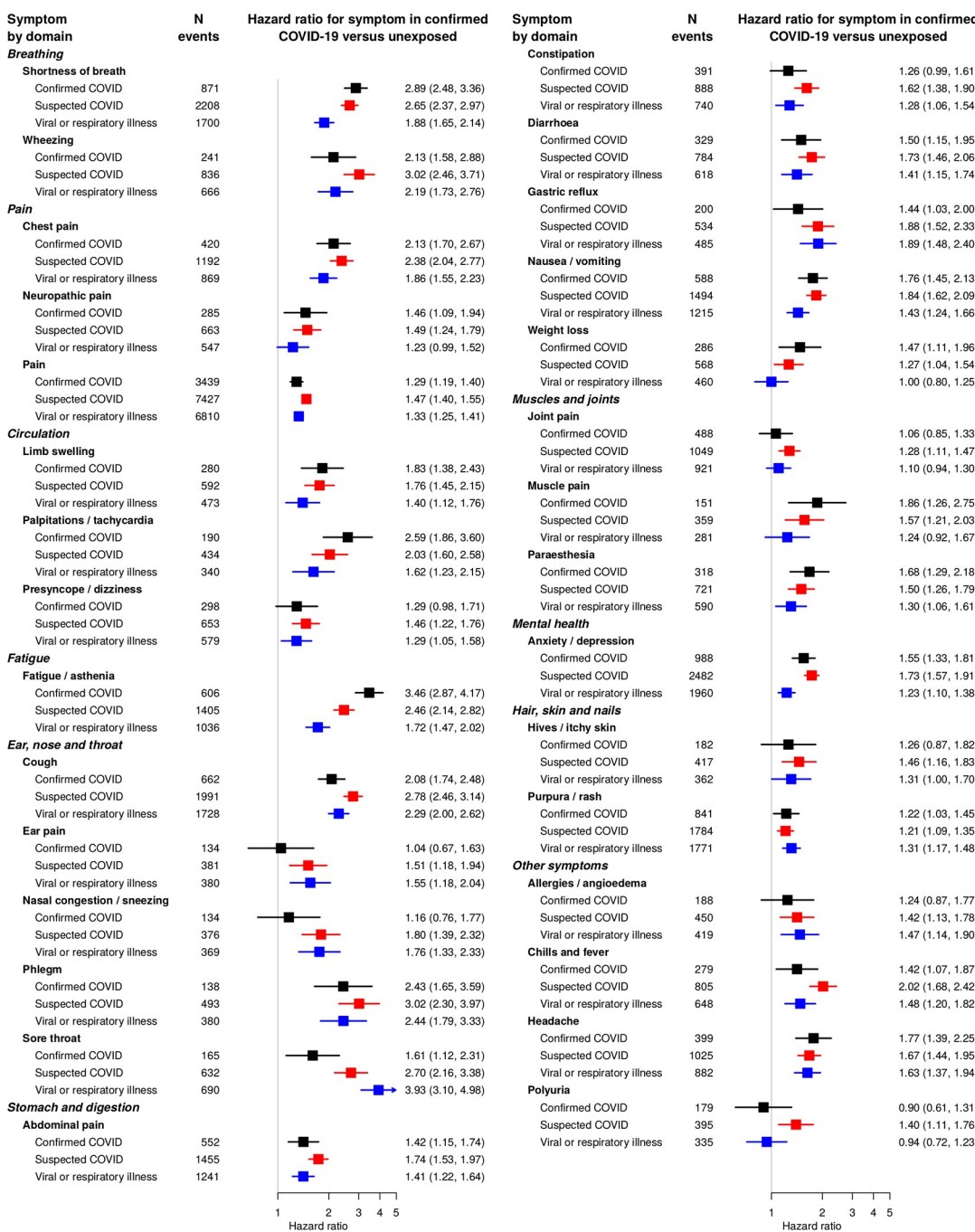

**Fig 6. Association of symptoms with prior suspected or confirmed COVID-19, or prior viral or respiratory illness.** Hazard ratios for association of symptoms with previous infection after 12 weeks, by case category. Hazard ratios were adjusted for age, sex, age/sex interaction, number of consultations in the year before the index date, number of symptom days 1–3 months before the index date, recording of the specific symptom 1–3 months before the index date, ethnicity, smoking, body mass index and a generated propensity score for acquiring COVID-19 infection, with the baseline hazard function stratified by general practice.

The following variables were associated with increased risk of presence of a WHO Long Covid symptom 12 or more weeks after COVID-19 infection in an unadjusted Cox model: female sex (HR 1.24, 95% CI 1.09, 1.41), age (HR 1.21 per 10 years older, 95% CI 1.17, 1.25), ex

**Table 2. Factors associated with Long Covid as defined by WHO symptoms or GP diagnosis of confirmed or suspected Long Covid.**

| Variable | Presence of WHO Long Covid symptom at least 12 weeks after COVID-19 infection | | GP diagnosis of Long Covid | |
| --- | --- | --- | --- | --- |
| | Crude | Adjusted | Crude | Adjusted |
| Female sex | 1.24 (1.09, 1.41) ** | 1.31 (1.15, 1.50) *** | 1.41 (0.92, 2.15) | 1.53 (0.98, 2.37) |
| Age (per 10 years) | 1.21 (1.17, 1.25) *** | 1.07 (1.03, 1.11) *** | 1.06 (0.95, 1.18) | 0.98 (0.87, 1.12) |
| Ethnicity (White as reference) | | | | |
| Black | 1.57 (0.94, 2.63) | 1.45 (0.86, 2.45) | 1.11 (0.15, 8.06) | 0.81 (0.11, 6.01) |
| South Asian | 0.94 (0.61, 1.46) | 0.94 (0.60, 1.47) | - | - |
| Mixed or Other | 0.65 (0.35, 1.21) | 0.60 (0.32, 1.12) | 1.47 (0.36, 6.08) | 1.12 (0.27, 4.73) |
| Missing ethnicity | 1.09 (0.96, 1.23) | 1.01 (0.88, 1.15) | 1.22 (0.82, 1.82) | 1.10 (0.71, 1.69) |
| Practice level deprivation quintile (least deprived as reference) | | | | |
| 2nd | 0.74 (0.59, 0.94) * | 0.89 (0.70, 1.13) | 0.60 (0.27, 1.35) | 0.74 (0.33, 1.69) |
| 3rd | 0.74 (0.60, 0.91) ** | 0.82 (0.67, 1.01) | 0.65 (0.33, 1.27) | 0.67 (0.34, 1.33) |
| 4th | 0.91 (0.76, 1.09) | 0.99 (0.83, 1.18) | 1.09 (0.64, 1.86) | 1.21 (0.70, 2.07) |
| 5th (most deprived) | 0.69 (0.58, 0.82) *** | 0.71 (0.59, 0.84) *** | 0.62 (0.35, 1.08) | 0.64 (0.36, 1.12) |
| Nation (England as reference) | | | | |
| Scotland | 0.63 (0.53, 0.74) *** | 0.77 (0.65, 0.92) ** | 0.60 (0.36, 1.02) | 0.64 (0.38, 1.11) |
| Wales | 0.83 (0.71, 0.98) * | 0.95 (0.80, 1.12) | 0.96 (0.57, 1.60) | 0.94 (0.55, 1.61) |
| Smoking status (never smoked as reference) | | | | |
| Ex smoker | 1.20 (1.05, 1.37) ** | 0.99 (0.87, 1.14) | 1.18 (0.79, 1.76) | 1.12 (0.74, 1.70) |
| Current smoker | 0.91 (0.74, 1.13) | 0.88 (0.71, 1.08) | 0.49 (0.21, 1.13) | 0.49 (0.21, 1.15) |
| Missing smoking status | 0.85 (0.61, 1.20) | 1.13 (0.79, 1.62) | 0.24 (0.03, 1.74) | 0.35 (0.05, 2.62) |
| BMI <18.5 | 1.09 (0.70, 1.71) | 0.88 (0.56, 1.37) | 0.50 (0.07, 3.67) | 0.49 (0.07, 3.63) |
| BMI 18.5–25 (reference) | | | | |
| BMI 25–30 | 1.00 (0.84, 1.19) | 0.94 (0.79, 1.12) | 0.74 (0.42, 1.29) | 0.70 (0.40, 1.23) |
| BMI 30–40 | 1.09 (0.92, 1.29) | 1.01 (0.85, 1.20) | 1.29 (0.78, 2.12) | 1.22 (0.73, 2.03) |
| BMI 40+ | 1.20 (0.93, 1.54) | 1.01 (0.79, 1.30) | 0.90 (0.39, 2.08) | 0.75 (0.32, 1.76) |
| Missing BMI | 0.63 (0.49, 0.81) *** | 0.76 (0.59, 0.99) * | 0.45 (0.18, 1.09) | 0.52 (0.21, 1.30) |
| Number of days with symptom mention 1–3 months before COVID-19 | 1.06 (1.05, 1.07) *** | 1.03 (1.02, 1.04) *** | 1.04 (1.01, 1.08) ** | 1.05 (1.01, 1.09) ** |
| Number of consultations in previous year | 1.03 (1.03, 1.04) *** | 1.02 (1.01, 1.02) *** | 1.00 (0.98, 1.02) | 0.97 (0.95, 1.00) * |
| Hospitalised during acute COVID-19 illness | 3.55 (3.12, 4.03) *** | 2.75 (2.39, 3.16) *** | 2.98 (1.99, 4.46) *** | 3.11 (1.99, 4.84) *** |

'Adjusted' hazard ratios are from a multivariable Cox model including all variables in this table. P values:

*** p < 0.001

** p < 0.01

* p < 0.05.

smoker (HR 1.20, 95% CI 1.05, 1.37), number of days with symptom recorded in 1–3 months prior to the index date (HR 1.06 per day of symptoms, 95% CI 1.05, 1.07), prior consultation frequency per year (HR 1.03, 95% CI 1.03, 1.04), and hospitalisation during the acute COVID-19 illness (HR 3.55, 95% CI 3.12, 4.03) (Table 2). Some factors were associated with reduced incidence of Long Covid as defined by WHO symptoms: practice-level deprivation (HR 0.69 for most compared to least deprived IMD quintile, 95% CI 0.58, 0.82) and residence in Scotland (HR 0.63, 95% CI 0.53, 0.74). On mutual adjustment, most of these associations remained statistically significant except being an ex smoker (Table 2). Using the GP diagnosis of suspected or confirmed Long Covid, the unadjusted hazard ratios for these variables were similar but the confidence intervals were wider (Table 2).

## Discussion

### Summary of main findings

By analysing primary care records including unstructured text from 60,800 patients across three UK nations during the early, pre-vaccination COVID-19 era, we demonstrate that a broad range of symptoms are associated with a history of COVID-19. However, some symptoms (e.g. gastrointestinal symptoms and anxiety) were common post COVID-19 but rarely associated with a Long Covid diagnosis. Patients were more likely to report symptoms of Long Covid or receive a Long Covid diagnosis if they were older, female, or hospitalised during their COVID-19 illness. The majority of symptom records were only available in the free text.

### Symptoms following COVID-19 diagnosis

Similar to previous studies using coded GP patient records [14] and longitudinal cohort studies [5, 6, 19], we found increased incidence of a wide range of symptoms in patients with a history of COVID-19. We found that fatigue was most strongly associated with a history of COVID-19 infection, consistent with previous studies [4]. The time period of our study (March to December 2020) meant that our study population was predominantly unvaccinated and the infections that occurred were with early variants of COVID-19. This should be taken into account when comparing our findings with those from more recent time periods.

The variety of clinical manifestations of Long Covid has led to the suggestion that there may be distinct subtypes of the disease, possibly with differing immunological mechanisms or other aspects of pathophysiology [33]. Our latent class analysis found that a two class model had the best fit with the data, consistent with longitudinal cohort analyses [34] and symptom tracker apps [17]. However, other clustering analysis have found different numbers of clusters [6–8, 14], but they have been carried out using different sets of clinical features (e.g., symptoms alone, or symptoms and diagnoses) and different cohorts (EHR cohorts, bespoke cohort studies, all patients with COVID-19 or only patients with a diagnosis of Long Covid). The UK primary care EHR study by Subramanian et al. [14] found three latent classes among non-hospitalised patients with a history of COVID-19. This may have been because this study was based solely on structured data (unlike ours), with fewer symptom records per patient, so the clusters may have been based on the most prominent symptom per patient.

### Diagnosis and risk factors for Long Covid

Consistent with prior literature, we found that increasing age, female sex [12, 16] and severity of acute COVID-19 [1] were associated with developing Long Covid. However, contrary to other studies, we found that socioeconomic deprivation was associated with a lower likelihood of a long Covid symptom or diagnosis being recorded. This may be due to inequality in access to care; perhaps patients registered in practices in more deprived areas were less able to access a GP, or the GP was less likely to record their symptoms or think about a Long Covid diagnosis. It is known that patients with long term somatic conditions with little evidence for underlying pathology may experience difficulty in obtaining a diagnosis [35], and it is probable that some patients with chronic symptoms following COVID-19 experienced similar difficulties.

Although a consensus definition of Long Covid exists, it is unknown how consistently it is applied in general practice, and associations need to be interpreted with caution. Variation in the diagnostic process means that the association between a condition and a Long Covid diagnosis may not be the same as the association with Long Covid itself.

Associations of patient characteristics with Long Covid will be determined most accurately from bespoke cohort studies; however such studies are typically not population based and

therefore cannot study symptomatology post Covid more broadly. Data linkages with secondary care, or a better means of sharing primary and secondary care diagnoses [36] may help.

In our study we found that no Long Covid diagnoses recorded using diagnosis codes. This was because the relevant SNOMED CT concepts (and linked Read terms used within GP systems) were not available for most of the study period. The UK SNOMED CT concept "Post COVID-19 syndrome" was released at the end of November 2020, and the International SNOMED CT concept "Chronic post-COVID-19 syndrome" was released at the end of January 2021. The ICD-10 code U09.9 "Post COVID-19 condition" was released in September 2020 [29]. Terminology updates in other coding systems were even more delayed; the US ICD-10-CM code for Long Covid was not available until October 2021 [9]. This shows the need for terminology systems to be updated in a timely manner to enable emerging conditions to be recorded faithfully.

More recent data does show that GPs are using clinical codes but the rate of coding is low, and varies between practices [23], so studies limited to coded data may still underestimate the incidence of GP diagnosed Long Covid.

## Limitations

The major strengths of our study were its population base, meaning that the results are likely to be generalisable, and the use of free text information to gather information about symptoms much more completely than coded data alone. However, our study has a number of limitations.

First, there was some uncertainty about the COVID-19 diagnosis itself and exposed or unexposed status of patients. This is because testing was not carried out systematically at the time of the study (before December 2020), so some patients diagnosed with COVID-19 might actually have had another diagnosis, and some 'unexposed' patients might have had asymptomatic COVID-19 infection, or not have sought healthcare for a COVID-like illness. To address this, we investigated associations among patients with different levels of certainty of COVID-19 (confirmed, suspected or possible), and verified that associations were stronger in groups that were more likely to have COVID-19 based on our definitions (Fig 6).

Second, free text analysis is always subject to error, because no computer algorithm can interpret the nuances of human language correctly all the time. Thus there may have been false negatives and false positives in reporting of symptoms, with a potential risk of bias due to misclassification. However, our manual review found that precision was over 85% with no significant difference between cases and controls. Therefore there may be some bias of hazard ratios towards the null, but there is unlikely to be any significant over-estimation of hazard ratios due to differential misclassification.

Third, there is likely to be variability in patients reporting symptoms to the GP, and the GP recording them in the clinical notes, and this may vary between COVID-19 and other illnesses. However, it should be noted that analyses limited to structured data have an additional risk of bias due to the GP's choice of which symptom(s) to record using clinical codes.

Fourth, we were unable to assess the severity of symptoms, and were therefore unable to fully apply the WHO diagnostic criteria for post COVID-19 condition [29]. Information on the functional level of patients, such as ability to work or perform daily activities of living, was not available.

Fifth, the time period of the study was limited, which means that we were unable to assess the effect of vaccination or different COVID-19 variants, and hazard ratio estimates for less common symptoms were imprecise. This was because of the governance requirements for analysing free text and the time limitation of the COPI notice, which expired on 30 June 2022.

Sixth, our data on social deprivation was at the practice rather than the patient level, so our estimates may have been affected by residual confounding by socio-economic factors, and we were unable to investigate whether patient-level deprivation affected the recording of symptoms and diagnoses by GPs.

## Recommendations for clinical care

We have identified a number of symptoms that are associated with a prior COVID infection but are less likely to be associated with a Long COVID diagnosis (e.g. gastrointestinal symptoms and anxiety). We suggest that clinicians bear in mind that such symptoms may constitute part of a Long Covid symptom cluster.

We recommend the accurate recording of symptom data, preferably in a structured way, in order to record and track a patient's disease over time and to facilitate research. While it is possible to analyse free text *post hoc*, as carried out in this study, it is difficult for algorithms to interpret complex contextual indicators. Semi-structured data entry systems (e.g. a 'history' box for patient symptoms) may help, and it is also important to improve the way that diagnosis information can be shared between healthcare settings [36].

## Recommendations for research

This study adds to the growing evidence of the value of free text analysis for healthcare research. Previous work on free text from primary care has demonstrated that symptoms are frequently not recorded in a structured way [24, 37]. Access to free text for clinical research in the UK is currently limited, even though it was vital for early work to validate coded GP diagnoses on which subsequent research depends [38]. This study had time-limited approvals, and a follow up study using more recent data could investigate the differences between COVID-19 variants and the impact of vaccination on post-Covid symptomatology. It would also be of interest to investigate the process of recording of diagnoses among GPs using structured and unstructured data.

Some large NHS trusts are building in-house infrastructure (such as the CogStack platform [39]) to analyse text in patient records. However, this is not feasible for general practices, which are too small to host such expertise and infrastructure themselves. There is a need for robust data governance arrangements to enable free text in medical records to be used for research in a safe, secure and timely manner [40]. A 'code to data' approach, as currently used for structured data in OpenSafely [23], may enable free text to be analysed securely with privacy protection. However, there will always be a need for samples of free text to be manually annotated to develop and validate the algorithms.

## Conclusion

Many symptoms are more common after COVID-19 infection, but only a few are commonly associated with a Long Covid diagnosis. There is a lack of structured recording of symptoms and Long Covid diagnoses in GP records, showing the importance of analysing free text in health records to study these topics.

## Supporting information

**S1 Text. Supplementary text.** Supplementary methods and results.
(PDF)

**S1 File. Analysis code.** Zip file containing analysis scripts.
(ZIP)

**S1 Table. Read terms for suspected or confirmed COVID-19.**
(CSV)

**S2 Table. Read terms for viral or respiratory illnesses.**
(CSV)

**S3 Table. Recording of symptoms in free text and coded data.** List of 98 symptoms investigated in this study, showing which symptoms are included in the WHO Long Covid case definition, which are used in the clustering analysis, and the percentage of days with either a free text or coded symptom record that the symptom is recorded using a coded entry.
(PDF)

**S4 Table. Two class latent class model for symptoms among patients with Long Covid.**
'Long Covid' was defined as either (a) presence of any symptom included in the WHO case definition of post COVID condition at least 12 weeks after the initial COVID-19 diagnosis, or (b) a symptom included in the secondary outcome of the study by Subramanian et al. (Nat Med 2022, doi: 10.1038/s41591-022-01909-w, S3 Table), for comparison with that study. For (a), symptoms in the 3 months after the WHO symptom were used in the latent class analysis; for (b) symptom records at any time were used. Cluster descriptions for (a): Class 1 (81.2%): Shortness of breath (30%), Fatigue / asthenia (23%), Anxiety / depression (23%), Cough (18%), Joint pain (11%). Class 2 (18.8%): Shortness of breath (51%), Nausea / vomiting (45%), Anxiety / depression (41%), Cough (39%), Abdominal pain (36%), Chest pain (32%), Fatigue / asthenia (30%), Diarrhoea (25%), Constipation (25%), Chills and fever (23%), Purpura / rash (21%), Wheezing (19%), Headache (17%), Phlegm (16%), Palpitations / tachycardia (16%), Gastric reflux (13%), Limb swelling (13%), Presyncope / dizziness (13%), Paraesthesia (12%), Weight loss (12%), Bloating (11%), Joint pain (10%). Cluster descriptions for (b): Class 1 (78.2%): Shortness of breath (18%), Anxiety / depression (17%), Purpura / rash (14%), Fatigue / asthenia (14%). Class 2 (21.7%): Shortness of breath (63%), Cough (55%), Fatigue / asthenia (40%), Anxiety / depression (40%), Nausea / vomiting (34%), Chest pain (33%), Abdominal pain (27%), Wheezing (23%), Diarrhoea (21%), Constipation (20%), Phlegm (20%), Purpura / rash (20%), Chills and fever (19%), Headache (19%), Palpitations / tachycardia (16%), Limb swelling (15%), Presyncope / dizziness (15%), Joint pain (14%), Paraesthesia (14%), Gastric reflux (13%), Weight loss (12%).
(PDF)

**S5 Table. Three class latent class model for symptoms among patients with Long Covid.**
'Long Covid' was defined as either (a) presence of any symptom included in the WHO case definition of post COVID condition at least 12 weeks after the initial COVID-19 diagnosis, or (b) a symptom included in the secondary outcome of the study by Subramanian et al. (Nat Med 2022, doi: 10.1038/s41591-022-01909-w, S3 Table), for comparison with that study. For (a), symptoms in the 3 months after the WHO symptom were used in the latent class analysis; for (b) symptom records at any time were used. Cluster descriptions for (a): Class 1 (50.8%): Anxiety / depression (26%), Fatigue / asthenia (19%), Abdominal pain (17%), Headache (13%), Nausea / vomiting (12%), Joint pain (11%), Diarrhoea (10%), Constipation (10%). Class 2 (35.2%): Shortness of breath (67%), Cough (34%), Fatigue / asthenia (28%), Chest pain (20%), Anxiety / depression (17%), Wheezing (13%). Class 3 (13.9%): Shortness of breath (64%), Anxiety / depression (49%), Nausea / vomiting (47%), Cough (42%), Chest pain (37%), Fatigue / asthenia (34%), Abdominal pain (34%), Constipation (28%), Diarrhoea (25%), Purpura / rash (24%), Wheezing (24%), Palpitations / tachycardia (22%), Chills and fever (21%), Headache (18%), Limb swelling (17%), Presyncope / dizziness (16%), Phlegm (16%), Gastric

reflux (14%), Weight loss (14%), Paraesthesia (13%), Joint pain (13%), Bloating (12%), Allergies / angioedema (11%). Cluster descriptions for (b): Class 1 (68.8%): Anxiety / depression (17%), Purpura / rash (15%), Fatigue / asthenia (12%), Nausea / vomiting (10%), Shortness of breath (10%). Class 2 (19.0%): Shortness of breath (74%), Cough (56%), Chest pain (29%), Fatigue / asthenia (27%), Wheezing (27%), Anxiety / depression (21%), Phlegm (16%). Class 3 (12.3%): Shortness of breath (53%), Nausea / vomiting (53%), Anxiety / depression (50%), Fatigue / asthenia (47%), Abdominal pain (43%), Cough (41%), Diarrhoea (32%), Constipation (31%), Chest pain (30%), Purpura / rash (29%), Headache (26%), Chills and fever (24%), Presyncope / dizziness (22%), Palpitations / tachycardia (21%), Paraesthesia (20%), Joint pain (18%), Weight loss (18%), Limb swelling (18%), Gastric reflux (18%), Bloating (15%), Wheezing (14%), Phlegm (12%).
(PDF)

**S1 Fig. Data flow diagram.**
(PDF)

**S2 Fig. Hazard ratios for all 89 symptoms.** Association of symptoms with previous COVID infection after 12 weeks. Hazard ratios were adjusted for age, sex, age/sex interaction, number of consultations in the year before the index date, number of symptom days 1–3 months before the index date, recording of the specific symptom 1–3 months before the index date, ethnicity, smoking, body mass index and a generated propensity score for acquiring COVID-19 infection, and stratified by general practice.
(PDF)

**S3 Fig. Hazard ratios by level of adjustment.** Association of symptoms with previous COVID infection after 12 weeks, by level of adjustment. 'Fully adjusted hazard ratios' were adjusted for age, sex, age/sex interaction, number of consultations in the year before the index date, number of symptom days 1–3 months before the index date, recording of the specific symptom 1–3 months before the index date, ethnicity, smoking, body mass index and a generated propensity score for acquiring COVID-19 infection, and stratified by general practice.
(PDF)

**S4 Fig. Hazard ratios by age.** Association of symptoms with previous infection after 12 weeks, by age. Hazard ratios were adjusted for age, sex, age/sex interaction, number of consultations in the year before the index date, number of symptom days 1–3 months before the index date, recording of the specific symptom 1–3 months before the index date, ethnicity, smoking, body mass index and a generated propensity score for acquiring COVID-19 infection, and stratified by general practice.
(PDF)

**S5 Fig. Hazard ratios by sex.** Association of symptoms with previous infection after 12 weeks, by sex. Hazard ratios were adjusted for age, sex, age/sex interaction, number of consultations in the year before the index date, number of symptom days 1–3 months before the index date, recording of the specific symptom 1–3 months before the index date, ethnicity, smoking, body mass index and a generated propensity score for acquiring COVID-19 infection, and stratified by general practice.
(PDF)

**S6 Fig. Hazard ratios by nation.** Association of symptoms with previous infection after 12 weeks, by UK nation. Hazard ratios were adjusted for age, sex, age/sex interaction, number of consultations in the year before the index date, number of symptom days 1–3 months before the index date, recording of the specific symptom 1–3 months before the index date, ethnicity,

smoking, body mass index and a generated propensity score for acquiring COVID-19 infection, and stratified by general practice.
(PDF)

**S7 Fig. Elbow plot for latent class analysis.** Elbow plot from latent class analysis for symptoms occurring within 3 months of a WHO Long Covid symptom, among patients at least 12 weeks after a COVID infection. The inflexion point at 2 classes shows that additional classes do not improve the fit of the model. AIC, Akaike Information Criterion; BIC, Bayesian Information Criterion; SABIC, sample size adjusted Bayesian Information Criterion; CAIC, Corrected Akaike Information Criterion.
(PDF)

## Acknowledgments

This study uses data from The Health Improvement Network (THIN) Database (a Cegedim Proprietary Database). This study uses patient information from NHS patients collected as part of their care and support. We acknowledge members of the THIN Advisory Committee lay panel and lay advisors with lived experience of Long Covid who have provided input throughout this project (Richard Ballerand, Beverley Chipp, Graeme Gosier, Sandra Jayacodi and Maneesh Juneja). We acknowledge the assistance of Dionisio Acosta Mena and Dennis Valentine from THIN Ltd. who extracted the data, ran the natural language processing software and manually anonymised the text samples. The views expressed are those of the authors and not necessarily those of the NIHR or the Department of Health and Social Care.

## Author Contributions

**Conceptualization:** Anoop D. Shah, Elizabeth Ford, Shamil Haroon, Valerie Kuan, Krishnarajah Nirantharakumar.

**Data curation:** Anoop D. Shah, Samir Dhalla.

**Formal analysis:** Anoop D. Shah, Anuradhaa Subramanian, Jadene Lewis, Shamil Haroon, Valerie Kuan, Krishnarajah Nirantharakumar.

**Resources:** Anuradhaa Subramanian, Samir Dhalla, Shamil Haroon, Valerie Kuan, Krishnarajah Nirantharakumar.

**Validation:** Anoop D. Shah, Jadene Lewis.

**Writing – original draft:** Anoop D. Shah.

**Writing – review & editing:** Anoop D. Shah, Anuradhaa Subramanian, Jadene Lewis, Samir Dhalla, Elizabeth Ford, Shamil Haroon, Valerie Kuan, Krishnarajah Nirantharakumar.

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
