## [Decision Letter · Decision Letter 0]

21 Jun 2023

PONE-D-23-14176Long Covid symptoms and diagnosis in primary care: a cohort study using structured and unstructured data in The Health Improvement Network primary care databasePLOS ONE

Dear Dr. Shah,

Thank you for submitting your manuscript to PLOS ONE. After careful consideration, we feel that it has merit but does not fully meet PLOS ONE’s publication criteria as it currently stands. Therefore, we invite you to submit a revised version of the manuscript that addresses the points raised during the review process.

We look forward to receiving your revised manuscript.

Kind regards,

Sreeram V. Ramagopalan

Academic Editor

PLOS ONE

Journal Requirements:

2. Please note that PLOS ONE has specific guidelines on code sharing for submissions in which author-generated code underpins the findings in the manuscript. In these cases, all author-generated code must be made available without restrictions upon publication of the work. 

Please review our guidelines at https://journals.plos.org/plosone/s/materials-and-software-sharing#loc-sharing-code and ensure that your code is shared in a way that follows best practice and facilitates reproducibility and reuse.

"This work was supported by Health Data Research UK, which receives its funding from the UK Medical Research Council, Engineering and Physical Sciences Research Council, Economic and Social Research Council, Department of Health and Social Care (England), Chief Scientist Office of the Scottish Government Health and Social Care Directorates, Health and Social Care Research and Development Division (Welsh Government), Public Health Agency (Northern Ireland), British Heart Foundation, and the Wellcome Trust. This study was supported by the National Institute for Health Research (NIHR) CONVALESCENCE grant (COV-LT-0009). ADS is funded by a postdoctoral fellowship from THIS Institute, NIHR (AI_AWARD01864 and COV-LT-0009), UKRI (Horizon Europe Guarantee for DataTools4Heart) and British Heart Foundation Accelerator Award (AA/18/6/24223). VK is supported by the UKRI/NIHR Strategic Priorities Award in Multimorbidity Research (MR/V033867/1) for the Multimorbidity Mechanism and Therapeutics Research Collaborative. EF is supported by the NIHR Applied Research Collaboration Kent Surrey and Sussex (grant number NIHR200179). KN has been awarded research grants from NIHR, UKRI/MRC, Kennedy Trust for Rheumatology Research, Health Data Research UK, Wellcome Trust, European Regional Development Fund, Institute for Global Innovation, Boehringer Ingelheim, Action Against Macular Degeneration Charity, Midlands Neuroscience Teaching and Development Funds, South Asian Health Foundation, Vifor Pharma, College of Police, and CSL Behring, all payments were made to his academic institution; Krishnarajah Nirantharakumar received consulting fees from BI, Sanofi, CEGEDIM, MSD and holds a leadership/fiduciary role with NICST, a charity and OpenClinical, a Social Enterprise."

"I have read the journal's policy and the authors of this manuscript have the following competing interests: ADS is funded by a postdoctoral fellowship from THIS Institute, NIHR (AI_AWARD01864 and COV-LT-0009), UKRI (Horizon Europe Guarantee for DataTools4Heart) and British Heart Foundation Accelerator Award (AA/18/6/24223). VK is supported by the UKRI/NIHR Strategic Priorities Award in Multimorbidity Research (MR/V033867/1) for the Multimorbidity Mechanism and Therapeutics Research Collaborative. EF is supported by the NIHR Applied Research Collaboration Kent Surrey and Sussex (grant number NIHR200179). KN has been awarded research grants from NIHR, UKRI/MRC, Kennedy Trust for Rheumatology Research, Health Data Research UK, Wellcome Trust, European Regional Development Fund, Institute for Global Innovation, Boehringer Ingelheim, Action Against Macular Degeneration Charity, Midlands Neuroscience Teaching and Development Funds, South Asian Health Foundation, Vifor Pharma, College of Police, and CSL Behring, all payments were made to his academic institution. KN received consulting fees from BI, Sanofi, CEGEDIM, MSD and holds a leadership/fiduciary role with NICST, a charity and OpenClinical, a Social Enterprise."

We note that you received funding from a commercial source: Boehringer Ingelheim, Vifor Pharma, CSL Behring, BI, Sanofi, CEGEDIM, MSD

Within this Competing Interests Statement, please confirm that this does not alter your adherence to all PLOS ONE policies on sharing data and materials by including the following statement: "This does not alter our adherence to PLOS ONE policies on sharing data and materials.” (as detailed online in our guide for authors http://journals.plos.org/plosone/s/competing-interests).  

If there are restrictions on sharing of data and/or materials, please state these. Please note that we cannot proceed with consideration of your article until this information has been declared. 

6. We noted in your submission details that a portion of your manuscript may have been presented or published elsewhere:

"Yes - a summary of this work has been presented as a digital poster at the ISPOR 2022 conference.

A preprint has also been published on medRxiv:

Long Covid symptoms and diagnosis in primary care: a cohort study using structured and unstructured data in The Health Improvement Network primary care database. medRxiv, version 1, PPR: PPR594314, DOI: 10.1101/2023.01.06.23284202"

Please clarify whether this conference proceeding or publication was peer-reviewed and formally published. If this work was previously peer-reviewed and published, in the cover letter please provide the reason that this work does not constitute dual publication and should be included in the current manuscript.

Reviewers' comments:

Reviewer's Responses to Questions

**Comments to the Author**

1. Is the manuscript technically sound, and do the data support the conclusions?

Reviewer #1: Partly

Reviewer #2: Yes

2. Has the statistical analysis been performed appropriately and rigorously? 

Reviewer #1: Yes

Reviewer #2: Yes

3. Have the authors made all data underlying the findings in their manuscript fully available?

Reviewer #1: No

Reviewer #2: No

4. Is the manuscript presented in an intelligible fashion and written in standard English?

Reviewer #1: Yes

Reviewer #2: Yes

5. Review Comments to the Author

Reviewer #1: The value of this paper is the agnostic approach using open text analysis, since long COVID is still a condition which is not well understood. I have some questions regarding the design and the available data:

- The authors talk about cohorts and have 4 exposure groups. It is confusing when these groups are named cases and controls which suggest that they are describing the outcomes (the symptoms).

- The cohorts should be more clearly defined, are they fixed or dynamic cohorts?

- Do all cohort members have prepandemic data on symptoms?

- Are the patients their own controls when comparing symptoms before and after infections?

- It seems that there was 1-to-1 matching between exposed (confusingly called cases) and unexposed patients for practice, age and sex. How is it then possible to estimate effects of sex and age, as is being done.

- Are cohort participants clustered within practices? If so, is there any intra-practice correlation of symptoms?

- How is index date defined?

- In other studies of long COVID, symptoms that has to do with cognitive function ("brain fog" etc) are often described. Are these type of symptoms not picked up by the techniques used here?

- It would be interesting if more data on the functional level of patients had been available, like ability to work or perform daily activities of living.

Reviewer #2: Shah et al describe an analysis of primary care EHR data to investigate symptoms associated with a diagnosis of long COVID. A particular strength of the study is the use of natural language processing to extract information from free text records that would otherwise be missed, and the work seems to provide a useful contribution to the literature (from my perspective as a statistician who has worked on COVID-19 studies). I have only relatively minor comments regarding the description of the study, along with some technical queries regarding the analysis.

As noted by previous reviewers, the dataset for analysis in this study is truncated at the end of December 2020. I do not think that this means that the analysis presented is no longer useful. However, given that the situation with long-COVID could now be quite different (given widespread vaccination and repeat infections with different variants etc.), I do feel that the timespan for the analysis needs to be specified in the Abstract and that the Introduction also needs to frame the analysis as effectively only considering the pre-vaccine period. These issues could also be explored in a little more depth in the Discussion.

I have some queries regarding the matching process:

-Firstly, were controls required to be ‘unexposed’ for the entire analysis period? I think that in an ideal nested case-control study design, control status would only be defined at the index date of the case (with follow-up censored in the event of subsequent infection). I don’t think that this should have had too large an impact for an analysis within this time period, but it would be a more important consideration for any analyses stretching into 2021 or 2022 given the very high cumulative incidence of SARS-CoV-2 at later time points.

-How precisely was age matched?

-I found it quite confusing to work out how the authors ended up with their final cohort of cases and controls. This is not clear at all in the text, but Figure 1 is helpful. I am not sure why exclusions based on index date, registration date or available follow-up would be conducted after the initial matching process – were these initial oversights? Or did this relate to data access constraints? I can see why reclassification of cases based on free text analysis would happen after the initial matching given the analysis required to process this information, but this is not explained clearly in the text.

-If the analysis had retained 1:1 matching, then I would have suggested consideration of a frailty term in the analysis models for matched pairs, but I don’t think that this is possible given the subsequent deviation from 1:1 matching. There is a related issue that the baseline hazard functions from index dates of controls don’t really have a clear interpretation given the deviation from 1:1 matching to cases, but the overall balance in calendar period of follow-up for cases and controls should be largely retained despite the reclassification of some cases and controls.

Further minor comments:

-It is stated that “We stratified hazard by general practice…”, I think that it would be correct and slightly clear to state “We stratified baseline hazard functions by general practice…”.

-“No patients had a coded diagnosis of post COVID condition (Long Covid)…”, I understand that the relevant ICD-10 code was not created until after the analysis period considered. This should perhaps be mentioned.

6. PLOS authors have the option to publish the peer review history of their article (what does this mean?). If published, this will include your full peer review and any attached files.

Reviewer #1: No

Reviewer #2: No

---

## [Author Response · Author response to Decision Letter 0]

9 Aug 2023

Response to reviewers: Long Covid symptoms and diagnosis in primary care: a cohort study using structured and unstructured data in The Health Improvement Network primary care database

August 2023

Reviewer #1: The value of this paper is the agnostic approach using open text analysis, since long COVID is still a condition which is not well understood. I have some questions regarding the design and the available data:

- The authors talk about cohorts and have 4 exposure groups. It is confusing when these groups are named cases and controls which suggest that they are describing the outcomes (the symptoms).

RESPONSE: We agree that the terminology is confusing. We have now removed the terms ‘cases’ and ‘controls’ and instead describe the exposure groups as four cohorts: ‘Confirmed COVID-19’, ‘Suspected COVID-19’, ‘Viral or respiratory illness’ and ‘Unexposed’. We have entirely rewritten the ‘Study population’ section of the methods. 

- The cohorts should be more clearly defined, are they fixed or dynamic cohorts?

RESPONSE: We have now stated in the text that these are fixed cohorts.

- Do all cohort members have prepandemic data on symptoms?

RESPONSE: Yes, all participants had to be registered with a participating practice for at least 1 year before the index date, and we expect that symptoms reported to the GP during that time would have been recorded.

- Are the patients their own controls when comparing symptoms before and after infections?

RESPONSE: No, we did not use the self controlled case series method (with each patient being their own control) because of temporal and seasonal changes which could influence symptom burden over time. This is why the primary comparison is between cohorts. However, prior symptom burden is used for adjustment in multivariable models.

- It seems that there was 1-to-1 matching between exposed (confusingly called cases) and unexposed patients for practice, age and sex. How is it then possible to estimate effects of sex and age, as is being done.

RESPONSE: Because of data restrictions, it was not possible to use the entire unexposed cohort and a selection procedure was therefore carried out for the unexposed cohort. Matching was used to generate an unexposed cohort with similar age and sex distribution to the exposed cohorts, and pseudo-diagnosis index dates that could be used for comparing symptoms over time. The matching was not used in analysis; regression models were unmatched and age and sex were included as covariates. This has now been explained more clearly in the methods, and the ‘study population’ section has been rewritten (see above).

- Are cohort participants clustered within practices? If so, is there any intra-practice correlation of symptoms?

RESPONSE: Our primary comparison was between exposed and unexposed patients. We assumed that recording practices (and therefore the baseline hazard of the symptom outcomes under investigation) may vary between practices, and wished to ensure that this did not affect the result. We therefore report analyses with the baseline hazard stratified by general practice.

- How is index date defined?

RESPONSE: The index date was the date of diagnosis of COVID-19 or a viral or respiratory illness for the exposed cohorts, and a pseudo-diagnosis date for unexposed patients. We have described the allocation of the index date in the ‘Study population’ section.

- In other studies of long COVID, symptoms that has to do with cognitive function ("brain fog" etc) are often described. Are these type of symptoms not picked up by the techniques used here?

RESPONSE: The natural language processing algorithm was designed to detect concepts in the text that currently exist in the Read terminology. The term ‘brain fog’ was not detected as it is not a Read term and had not been programmed into the model. Future NLP models can be trained to recognise this and other novel or non-standard symptom descriptions.

- It would be interesting if more data on the functional level of patients had been available, like ability to work or perform daily activities of living.

RESPONSE: We agree this would be ideal. We have added this sentence to the fourth limitation in the discussion:

Information on the functional level of patients, such as ability to work or perform daily activities of living, was not available.

Reviewer #2: Shah et al describe an analysis of primary care EHR data to investigate symptoms associated with a diagnosis of long COVID. A particular strength of the study is the use of natural language processing to extract information from free text records that would otherwise be missed, and the work seems to provide a useful contribution to the literature (from my perspective as a statistician who has worked on COVID-19 studies). I have only relatively minor comments regarding the description of the study, along with some technical queries regarding the analysis.

As noted by previous reviewers, the dataset for analysis in this study is truncated at the end of December 2020. I do not think that this means that the analysis presented is no longer useful. However, given that the situation with long-COVID could now be quite different (given widespread vaccination and repeat infections with different variants etc.), I do feel that the timespan for the analysis needs to be specified in the Abstract and that the Introduction also needs to frame the analysis as effectively only considering the pre-vaccine period. These issues could also be explored in a little more depth in the Discussion.

RESPONSE: We have now mentioned the timespan of the study in the abstract and introduction. We have modified the Abstract methods to say:

We used primary care electronic health record data until the end of December 2020 ...

RESPONSE: We have inserted this sentence into the last paragraph of the introduction:

Our study is based on data to the end of December 2020, i.e. the first wave of COVID-19 infections in a predominantly unvaccinated population.

RESPONSE: In the first sentence of the discussion, we now mention that this is a pre-vaccination population:

By analysing primary care records including unstructured text from 60,800 patients across three UK nations during the early, pre-vaccination COVID-19 era, …

RESPONSE: We have added the following sentence to the discussion section: symptoms following COVID-19 diagnosis:

The time period of our study (March to December 2020) meant that our study population was predominantly unvaccinated and the infections that occurred were with early variants of COVID-19. This should be taken into account when comparing our findings with those from more recent time periods.

I have some queries regarding the matching process:

-Firstly, were controls required to be ‘unexposed’ for the entire analysis period? I think that in an ideal nested case-control study design, control status would only be defined at the index date of the case (with follow-up censored in the event of subsequent infection). I don’t think that this should have had too large an impact for an analysis within this time period, but it would be a more important consideration for any analyses stretching into 2021 or 2022 given the very high cumulative incidence of SARS-CoV-2 at later time points.

RESPONSE: We have now mentioned the timespan of the study in the abstract and introduction. Unexposed patients were required to be unexposed for the entire duration of the analysis period. RESPONSE: We agree that different analysis methods will need to be used for analyses in later years as the cumulative incidence of history of COVID-19 infection will be very high.

-How precisely was age matched?

RESPONSE: We aimed to match cohort distributions rather than carry out a matched analysis, but because of restrictions on data access it was most feasible to do this via preliminary patient level matching and then applying further eligibility criteria as per Figure 1. In the initial patient level matching, patients were matched on age (+/- 3 years), sex and practice. The original text also stated erroneously that patients were matched on ethnicity; this was originally intended in our analytic protocol, but was not carried out as there was a considerable amount of missing data and some ethic groups had small numbers. We have corrected the text.

-I found it quite confusing to work out how the authors ended up with their final cohort of cases and controls. This is not clear at all in the text, but Figure 1 is helpful. I am not sure why exclusions based on index date, registration date or available follow-up would be conducted after the initial matching process – were these initial oversights? Or did this relate to data access constraints? I can see why reclassification of cases based on free text analysis would happen after the initial matching given the analysis required to process this information, but this is not explained clearly in the text.

RESPONSE: Yes, this is correct, there were data access constraints which meant that it was not possible to obtain the free text until the cohorts had been provisionally defined based on coded data, and then additional exclusions were carried out subsequently when the data were accessible to researchers. This is now explained more clearly in the “Study population” section of the methods (see response to reviewer 1 above).

-If the analysis had retained 1:1 matching, then I would have suggested consideration of a frailty term in the analysis models for matched pairs, but I don’t think that this is possible given the subsequent deviation from 1:1 matching. There is a related issue that the baseline hazard functions from index dates of controls don’t really have a clear interpretation given the deviation from 1:1 matching to cases, but the overall balance in calendar period of follow-up for cases and controls should be largely retained despite the reclassification of some cases and controls.

RESPONSE: We apologise for the confusion in the descriptions. The cohorts are not considered to be matched for analysis. The unexposed cohort is effectively a stratified random sample intended to have a similar distribution of demographic characteristics to the exposed cohort.

Further minor comments:

-It is stated that “We stratified hazard by general practice…”, I think that it would be correct and slightly clear to state “We stratified baseline hazard functions by general practice…”.

RESPONSE: Thank you for this comment, we have modified the text and the figure captions.

-“No patients had a coded diagnosis of post COVID condition (Long Covid)…”, I understand that the relevant ICD-10 code was not created until after the analysis period considered. This should perhaps be mentioned.

RESPONSE: Thank you for this comment. We have expanded on the delays in availability of terminology concepts in the discussion.

---

## [Editor Report · Decision Letter 1]

11 Aug 2023

Long Covid symptoms and diagnosis in primary care: a cohort study using structured and unstructured data in The Health Improvement Network primary care database

PONE-D-23-14176R1

Dear Dr. Shah,

We’re pleased to inform you that your manuscript has been judged scientifically suitable for publication and will be formally accepted for publication once it meets all outstanding technical requirements.

Kind regards,

Sreeram V. Ramagopalan

Academic Editor

PLOS ONE
---

## [Editor Report · Acceptance letter]

18 Sep 2023

PONE-D-23-14176R1 

Long Covid symptoms and diagnosis in primary care: a cohort study using structured and unstructured data in The Health Improvement Network primary care database 

Dear Dr. Shah:

I'm pleased to inform you that your manuscript has been deemed suitable for publication in PLOS ONE. Congratulations! Your manuscript is now with our production department. 

Kind regards, 

on behalf of

Dr. Sreeram V. Ramagopalan 

Academic Editor

PLOS ONE